# 3D-ViT-UNet: 3D Vision transformer based Unet-like model for Volumetric Brain Tumor Segmentation

Sikandar Afridi[1]⊙, Atif Jan[1]⊙, Muhammad Abeer Irfan[2]*⊙, Muhammad Irfan Khattak[1]⊙, Taimur Ahmed Khan[1]⊙

**1** Department of Electrical Engineering, University of Engineering and Technology, Peshawar, Pakistan,
**2** School of Computer Science, Technological University Dublin (TU Dublin), Dublin, Ireland

⊙ These authors contributed equally to this work.
* Muhammad.Irfan2@TUDublin.ie

## Abstract

Accurate volumetric segmentation of 3D medical imaging modalities is critical for therapy planning and clinical diagnosis, particularly for brain tumor delineation. Traditional convolutional neural network (CNN)-based architectures face challenges while capturing global contextual information and modeling long-range dependencies in complex 3D volumetric data, limiting their segmentation performance. Transformer-based models have emerged as promising alternatives to CNNs for such tasks, addressing their limitations in capturing global spatial dependencies. We propose 3D-ViT-UNet, a novel U-shaped vision transformer (ViT)-based encoder-decoder architecture for end-to-end volumetric brain tumor segmentation. The model employs 3D Window Multi-Head Self-Attention (3D-W-MSA) to capture local features and a 3D Dilated-Window Multi-Head Self-Attention (3D-DW-MSA) to capture global features while reducing computational complexity. Moreover, for preserving absolute and relative positional information and preventing permutation equivalence limitation in transformers, a dynamic position encoding strategy is integrated. The proposed model demonstrates state-of-the-art (SOTA) performance for brain tumor segmentation on the BraTS 2020 dataset. It achieves a superior average Dice Similarity Coefficient (DSC) of 84.81% and a Hausdorff Distance (HD) of 4.87 mm with reduced computational complexity compared to existing methods. Also, an improvement in delineation of tumor boundaries and accurate segmentation across modalities is demonstrated through the qualitative results. Extensive quantitative and qualitative evaluations highlight the capability of 3D-ViT-UNet to achieve high accuracy with a smaller model size and lower FLOPs, making it an effective and efficient solution for clinical applications involving volumetric brain tumor segmentation.

**Data availability statement:** The data used in this study are publicly available on Kaggle. Specifically, the volumetric brain tumor MRI data were obtained from the "Brain Tumor Segmentation (BraTS) Dataset" hosted on Kaggle and can be accessed at https://www.kaggle.com/datasets/awsaf49/brats20-data-set-training-validation All preprocessing scripts, model architecture implementations, training code, and evaluation scripts for the proposed 3D-ViT-UNet model are publicly available in a GitHub repository at https://github.com/maikhan/3D_ViT_Unet.

**Funding:** The author(s) received no specific funding for this work.

**Competing interests:** The authors have declared that no competing interests exist.

## Author summary

Brain tumors have varying sizes and shapes across MRIs; therefore, their accurate volumetric segmentation is a challenging task before therapies and surgeries. It is a time-consuming manual task, and results can differ between experts. We present 3D-ViT-UNet, an end-to-end volumetric segmentation model that processes an MRI as a volume rather than independent slices. Our design combines two attention mechanisms: 3D window attention to capture fine local structure and 3D dilated-window attention to efficiently capture broader context for full tumor extent. To keep the correct spatial order of input 3D patches, we add a dynamic, input-dependent position encoding that adapts to each MRI scan. Our method achieved state-of-the-art performance with a DSC of 84.81% and an average HD95 of 4.87 mm on the BraTS 2020 dataset. This confirms that 3D-ViT-UNet is an effective and efficient solution for clinical applications, providing high segmentation accuracy with a smaller model size and reduced computational cost.

## 1. Introduction

Medical image segmentation is crucial for medical diagnosis, therapy planning, patient monitoring, and treatment across diseases, including brain tumors. Precise delineation of tumor boundaries allows efficient therapy planning, surgical interventions, and monitoring of disease progression. Medical image segmentation extracts features of anatomical structures and regions of interest from imaging modalities, including organs and their boundaries [1]. Volumetric segmentation of brain tumors is more challenging due to the complex and heterogeneous appearances of 3D MRIs, demanding reliable and precise models. Recently, enhancements in deep learning (DL) and computer vision have remarkably improved performance in segmentation applications, including brain tumor [2], cardiovascular [3], and liver [4] segmentation.

Over the years, CNNs are increasingly being adopted as standards for DL frameworks in medical image analysis [5], excelling in classification and recognition as well as segmentation of images [6], [7]. Their strong feature extraction capability and precision make them highly reliable for segmentation of medical images [8], [9]. Few of the most prominent CNN-based encoder–decoder models with skip connections for brain tumor segmentation include HPU-Net [10], SegResNet [11], and Res-UNet [12]. A U-shaped CNN-based encoder-decoder framework with skip connections known as U-Net is one of the most promising CNN-based methods used for brain tumor segmentation, one of the most challenging tasks in medical image analysis [13,14]. U-Net has shown promising results for a variety of segmentation tasks, including tumor, cardiac, and lesion segmentation, as well as organ segmentation, such as liver, kidney, brain, and lungs [15,16]. U-Nets have established themselves as a cornerstone of medical image segmentation, particularly for brain tumor segmentation, due to their architectural innovation and development. U-Net++ [17], S3DUNet [18],

UNet3+ [19], and 3D U-Net [13] are various U-Net models proposed for segmenting medical image modalities, including MRI, CT, and X-rays.

Unlike 2D methods, volumetric segmentation demonstrates superior precision as it processes the entire 3D volume and captures spatial contextual information across slices [20]. Models such as 3D U-Net [13] and Non-Local U-Net [21] have been developed to address the challenges in segmentation of 3D medical modalities. However, CNN-based methods like U-Net remain sub-optimal in terms of performance for volumetric brain tumor segmentation due to their inherited limitations, such as limited kernel size, which restricts modeling long-range dependency and capturing global context [22].

ViTs use self-attention to capture global contextual information and model long-range dependencies, allowing them to accurately identify complex structures and patterns in medical modalities. This gives them an advantage over CNNs like U-Net [23], which enhances feature extraction and pattern recognition capabilities [22,23], resulting in superior performance for volumetric brain tumor segmentation.

To address the challenging medical image segmentation task, several ViT-based models have been designed. UNETR [24] replaced the convolutional-based encoder with the ViT-based encoder, followed by models like TransUnet [22], TransBTS [25], and TransFuse [26] that advanced multi-organ and brain tumor segmentation. Recently, medical image segmentation models such as MISSFormer [27], Swin-Unet [28], and TransDeepLab [29] have employed pure transformer encoder–decoder structures to capture richer features. Subsequently, UNETR++ [30], introduced as an extension of UNETR [24] with paired-attention and TMA-TransBTS [31] employed multi-scale self-attention and cross-attention to improve segmentation accuracy.

Although promising, the above-mentioned ViT-based methods still have several significant limitations, affecting their segmentation performance. These limitations include:

- 2D-based Approaches for 3D Data: Most ViT-based models are designed for 2D medical image segmentation [22,26–29]. These models process 3D medical images, such as 3D MRIs, by dividing them into 2D slices and applying 2D operations and layers to each slice. These approaches lead to a loss of spatial contextual information, which is crucial for accurate volumetric segmentation.

- Loss of Positional Information: A loss of patch order occurs in the input sequence due to the permutation-based operation of the self-attention mechanism. It degrades the performance of the models, as segmentation process is spatial position depended. Although strategies like absolute and relative position encoding [32,33] have been proposed to address this issue, they often fail to generalize for different image sizes due to fixed patch constraints. Furthermore, these strategies overlook the significance of absolute positional information, critical for medical image segmentation.

- High Computational Cost of 3D Models: 3D medical image segmentation is often large and computationally costly, leading to the development of methods that aim to reduce operational costs [34,35]. However, these methods face limitations in achieving sufficient global information fusion. Therefore, designing a computationally efficient ViT-based method that retains voxel-level information and preserves interdependencies across arbitrary positions in large 3D tensor inputs is a crucial challenge for volumetric brain tumor segmentation.

Motivated by the pure transformer-based U-shape encoder-decoder architecture of Swin-Unet [28] and the dilated convolutional kernels proposed by [36,37], a novel pure 3D ViT-based U-Net-like model, 3D-ViT-U-Net is presented for volumetric brain tumor segmentation. The model is designed to harness both local and global semantic features utilizing an enhanced encoder-decoder architecture with skip connections. The proposed 3D-ViT-U-Net processes tokenized patches of volumetric images, which are fed into a 3D Video Swin Transformer [38] utilizing 3D Window Multi-Head Self-Attention mechanism (3D-W-MSA) and 3D Dilated-Window Multi-Head Self-Attention (3D-DW-MSA). These replace conventional 3D Shifted-Window Multi-Head Self-Attention (3D-SW-MSA) to improve feature extraction capabilities. 3D-W-MSA captures local features, while 3D-DW-MSA, in conjunction with a dilation mechanism, enables efficient global

feature extraction. This approach expands the receptive field without increasing the number of patches, reducing computational costs. Furthermore, 3D-W-MSA integrates patch position encoding using depth-wise convolution [39] to address permutation invariance issues [40], further enhancing local self-attention. A symmetrically designed decoder reconstructs feature maps, facilitating accurate segmentation. The key contributions of our work are as follows:

1. A 3D ViT-based U-shaped architecture is proposed for end-to-end volumetric brain tumor segmentation. This architecture processes entire 3D medical modalities directly, preserving spatial contextual information throughout segmentation.

2. A dynamic input-dependent position encoding strategy is used to compute absolute and relative patch positions, preserving the sequential order of input patches and enhancing spatial coherence.

3. A 3D Dilated-Window Multi-Head Self-Attention with a dilation mechanism is designed to exploit the extraction of global semantic features. This approach enables efficient self-attention computation by expanding the receptive field without increasing the number of patches, thus reducing computational costs.

## 2. Related work

### 2.1. CNN-based models for medical image segmentation

CNN-based models, particularly U-Nets, have been highly effective for medical image segmentation, most notably U-Net models including U-Net++ [17], S3DUNet [18], UNet3+ [19], and 3D U-Net [13]. Methods such as large kernel sizes [41], pyramid pooling [42], and dilated convolution [36] have been proposed to address the issue of locality in CNN-based models. Dilated convolution has achieved better results for medical images [37]. Therefore, we have employed a dilation mechanism within the self-attention mechanism of the proposed model to extract global semantic features.

### 2.2. Adoption of transformer in computer vision

Primarily developed to perform tasks related to natural language processing (NLP) [40], other state-of-the-art (SOTA) models have been outperformed by transformers for applications like machine translation, large language models, multimodal language models, and sequence-to-sequence models. Their adaptability to capture global contextual information across medical modalities has resulted in widespread use for medical image analysis and segmentation, where self-attention mechanisms are integrated with CNN-based models. For example, non-local operations [43] and additive gated attention modules in skip connections [44] have been incorporated into neural networks and U-shaped encoder-decoder architecture. Over time, transformers have been adapted for computer vision. Initially, self-attention was applied locally to each pixel's neighborhood rather than to the entire global context [45]. The sparse transformers were introduced to further extend this approach by applying scalable approximations to compute global self-attention [46]. Recently introduced ViT has become the first to apply transformers to compute global self-attention for a full-sized image, leading to remarkable image classification and segmentation performance, surpassing traditional SOTA methods [23]. These advances underscore the potential of transformer-based methods to perform comprehensive and effective tasks in computer vision.

### 2.3. ViT-based models for medical image segmentation

leveraging the architectural innovations and strengths, various ViT-based methods have been proposed to perform the challenging task of medical image segmentation. The earliest ViT-based architectures for medical image segmentation that utilize ViT as an encoder with a CNN-based decoder capable of modeling long-range dependencies include TransUnet [22], UNETR [24], and UCTransNet [47]. This combination harnesses the capabilities of CNNs and transformers to capture global contextual information and localization precision, respectively. In TransBTS [25], the extracted feature maps by a CNN-based encoder are refined using a ViT to capture global contextual information. LeViT-UNet [48],

presented a U-Net where a lightweight ViT encoder is integrated to balance segmentation performance with computational efficiency for several medical image modalities. TransFuse [26] addresses the issue of inadequate feature fusion in traditional architectures by incorporating a fusion block that fuses the features extracted by the CNN block with the transformer components. Similarly, CoTr [49] presents the integration of CNN and transformer blocks with a cross-attention mechanism, which captures the local as well as global context, enabling more accurate medical image segmentation. To improve the segmentation performance of ViT for multimodal medical imaging analysis, SwinCross introduces a cross-modal Swin Transformer encoder [50]. A CNN-based decoder is paired with a cross-modal attention module for CT and PET modalities. UNETR++ [30], a hybrid model built on UNETR for medical image segmentation, leverages efficient paired-attention to capture spatial context. An extended version of TransBTS known as TMA-TransBTS [31] for volumetric brain tumor segmentation employs a token-mixing attention mechanism for better multi-scale feature representation.

### 2.4. ViT-based U-shaped architectures for medical image segmentation

Due to the advancement in transformer-based methods and SOTA performance of encoder-decoder architectures with a U-shape for medical image segmentation, many transformer-based U-shaped hierarchical models with W-MSA and SW-MSA have been designed to capture both local and global contextual attention. Missformer [27], hybrid U-shaped ViT-based architecture, combines self-attention mechanisms with traditional convolutions to address the challenge of multi-scale feature extraction. Similarly, nnFormer [51], with a hybrid U-shaped 3D transformer-based model, demonstrates a significant performance improvement over CNN-based models across different medical imaging modalities. Swin-Unet [28], a pure ViT-based model consisting of Swin Transformer blocks with a U-Net-like architecture, uses SW-MSA to capture multi-scale features while reducing computational complexity and achieving high segmentation accuracy.

Despite recent progress, none of the above-mentioned models for brain tumor segmentation, i.e., UNETR [24], TransBTS [25], nnFormer [51], and Swin-Unet [28], is a pure 3D ViT-based architecture. Unlike these approaches, our aim is to explore the potential of a pure 3D ViT as an encoder as well as a decoder in a U-shaped architecture to perform end-to-end volumetric brain tumor segmentation.

## 3. Methodology

This work proposes a SOTA approach to address the most prominent challenges of volumetric brain tumor segmentation with a ViT-based U-shaped model. Our model, termed 3D-ViT-UNet, leverages the strengths of ViTs to model long-range dependencies and capture global contextual information while maintaining the architectural advantages of U-Net. Designed specifically for medical imaging, this architecture is optimized to process entire 3D medical volumes, preserving spatial context and enhancing segmentation accuracy.

### 3.1. Architecture overview

An overview of the proposed 3D-ViT-UNet, a U-shaped model, is presented in Fig 1. The U-Net-like architecture includes three fundamental components: encoder, bottleneck, and decoder, along with skip connections. 3D-ViT-UNet encoder includes a 3D patch embedding layer used to project 3D images into sequences, followed by three 3D-ViT blocks for feature extraction. Each 3D-ViT block is succeeded by a layer of down-sampling, producing a hierarchical feature map of the 3D input. The bottleneck consists of two successive 3D-ViT blocks to learn deeper feature representations. Symmetrical to the encoder, the decoder contains three 3D-ViT blocks. Each 3D-ViT block is followed by an up-sampling layer, with a final 3D patch expanding layer. This component primarily expands the encoder-generated feature maps with low-resolution to make mask predictions. As in other U-shaped architectures, skip connections are employed to concatenate the encoder feature maps with the corresponding decoder feature maps, supplementing fine-grained spatial detail that may be lost during down-sampling.

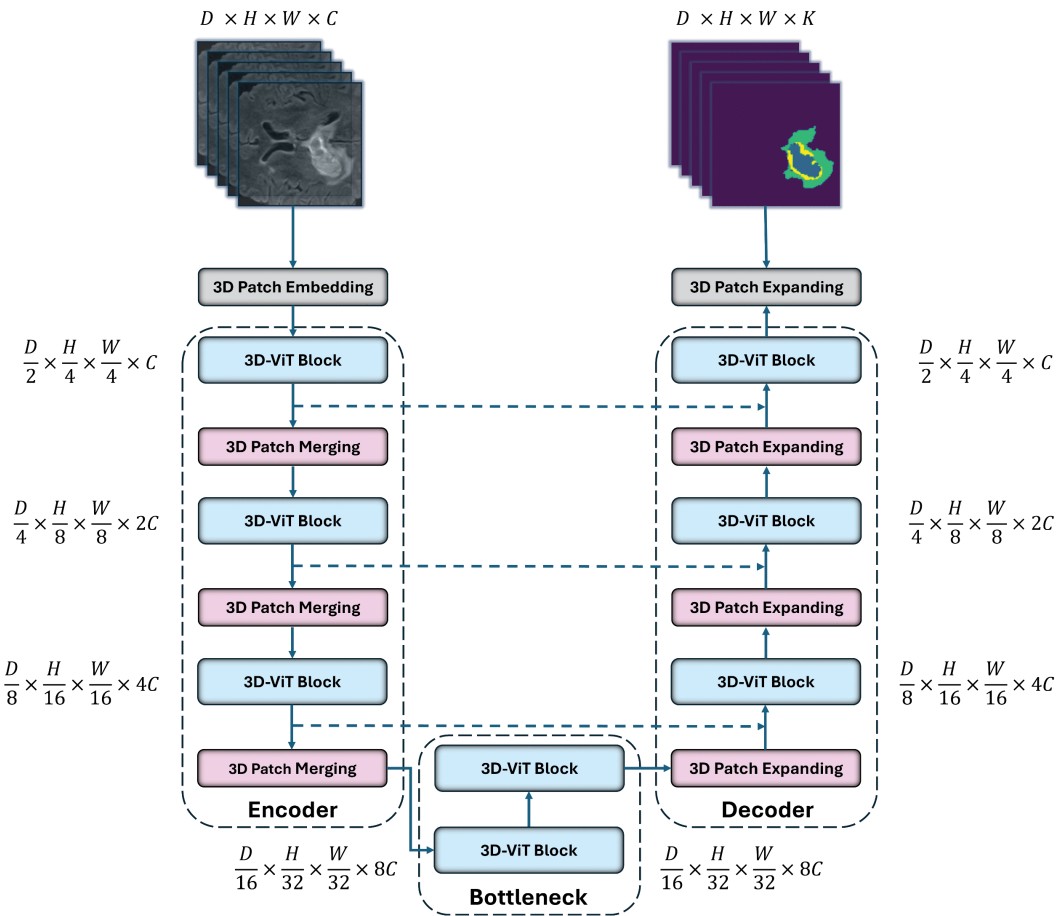

**Fig 1. Architectural overview of 3D-ViT-UNet.** Each 3D-ViT block consists of a DIPE, 3D-W-MSA, and 3D-DW-MSA. Input size of the $k^{th+1}$ 3D-ViT block is the same as the corresponding output size of the $k^{th}$ 3D-ViT block. The numbers of patches are represented in the round brackets, i.e., $(\frac{D}{4} \times \frac{H}{8} \times \frac{W}{8} \times 2C)$, $(\frac{D}{8} \times \frac{H}{16} \times \frac{W}{16} \times 4C)$ and $(\frac{D}{16} \times \frac{H}{32} \times \frac{W}{32} \times 8C)$, which are one-dimensional when computed in Transformers.

## 3.2. 3D-ViT block

Current ViT-based volumetric segmentation methods lack robust [24] and efficient mechanisms like 3D-DW-MSA [22], to manage computational costs while capturing global contextual information in large and complex datasets like 3D brain MRIs. Motivated by [36] and [37], the 3D-SW-MSA in the video Swin Transformer [38] is replaced with 3D-DW-MSA in our proposed 3D-ViT block. This modification reduces computational cost while maintaining the model's capability of capturing global contextual information.

### 3.2.1. Dynamic input-dependent position encoding (DIPE).
DIPE adds positional information to enhance feature maps in the 3D-ViT block by adding spatial context, allowing the model to leverage spatial dependencies. Contrary to conventional fixed position encoding methods [32,33], DIPE uses a 3D depth-wise convolution (3D-DW-Conv) [39] to learn positional information adaptively from a 3D input medical image. Spatial information is embedded directly into feature maps by DIPE in order to address the lack of positional awareness in the self-attention mechanism crucial for volumetric segmentation of medical images.

In order to reduce computational cost, one convolutional filter is employed per channel at the input of 3D-DW-Conv, unlike traditional convolution, where a filter is applied to all channels. In the 3D-ViT block, positional information is learned by applying 3D-DW-Conv, which is then added to the feature map at the input denoted as $X \in \mathbb{R}^{B \times C \times D \times H \times W}$:

$$X' = \text{DW-}Conv(Resize(X)), \tag{1}$$

$$X_{pos} = Resize(X') + X. \tag{2}$$

Here, $B$ denotes the batch size, $C$ refers to the number of channels, and $D$, $H$, $W$ represent the depth, height, and width of the feature map, respectively. $X_{pos} \in \mathbb{R}^{B \times C \times D \times H \times W}$ represents the position-encoded feature map, containing dynamic positional information. The resulting feature maps are reshaped to the required dimension of DW-Conv, followed by the extraction of patch position information. DIPE enhances the model's ability of capturing the spatial relationship accurately in complex volumetric modalities, such as brain MRIs, where all three dimensions are critical. The model provides reliable and accurate segmentation results while maintaining spatial coherence via embedded positional information. Moreover, DIPE allows the model to process complex anatomical scales and structures in a generic manner [52], providing a prominent advantage in accurate and precise 3D medical image segmentation.

### 3.2.2. 3D Window Multi-Head Self-Attention (3D-W-MSA).

3D-W-MSA is one of the main components of our proposed 3D-ViT block, where contextual information is acquired locally from the input feature map. Conventional self-attention methods, such as vanilla transformers, find long-range dependencies by computing pairwise relationships between patches at a high computational cost. However, 3D-W-MSA performs limited self-attention operation to non-overlapping local windows referred to as 3D-window. It captures the important contextual information within each local 3D-window with more computational efficiency. 3D-W-MSA proves to be a better self-attention mechanism, both in terms of accuracy as well as computational efficiency for 3D medical image segmentation with complex patterns, such as volumetric brain tumor segmentation.

In 3D-W-MSA, non-overlapping 3D windows of equal sizes are formed by dividing a given feature map, then for every window, self-attention is computed as depicted in Fig 2. In a 3D-window, the number of patches $N$ is calculated as $N = p_D \times p_H \times p_W$, where $p_D$, $p_H$, and $p_W$ represent the number of patches in a 3D-window along the $D$, $H$, and $W$ dimensions of the feature map, respectively. The query vector, key vector, and value vector, denoted by $Q$, $K$ and $V$, respectively, are obtained by projecting the $X_{pos}$, linearly as:

$$Q, K, V = \text{Linear}(X_{pos}). \tag{3}$$

Where $Q$, $K$ and $V \in \mathbb{R}^{B \times N \times d}$, with $d$ representing the dimension of each of these vectors. As in [40], for each 3D-window, self-attention for $Q$, $K$ and $V$ is computed as:

$$\text{Attention}(Q, K, V) = \text{Softmax}\left(\frac{QK^T}{\sqrt{d}} + \mathbf{B}^{\text{rel}}\right) V. \tag{4}$$

Here, $B^{\text{rel}} \in \mathbb{R}^N$ denotes the relative position bias, which enables the model to understand the positional information within a given 3D-window.

Compared to the quadratic computational complexity of conventional methods, 3D-W-MSA reduces the complexity to be linearly related to the total number of patches for an entire feature map. Denoted by ($\mathscr{C}$), the computational complexity is computed as:

$$\mathscr{C}(\text{MSA}) = 4\Omega C^2 + 2\Omega^2 C. \tag{5}$$

$$\mathscr{C}(\text{3D-W-MSA}) = 4\Omega^2 + 2(N)(\Omega)C. \tag{6}$$

Where $\Omega = D \times H \times W$ refers to the total number of patches in the feature map. Normally, $N$ is quite less than $\Omega$.

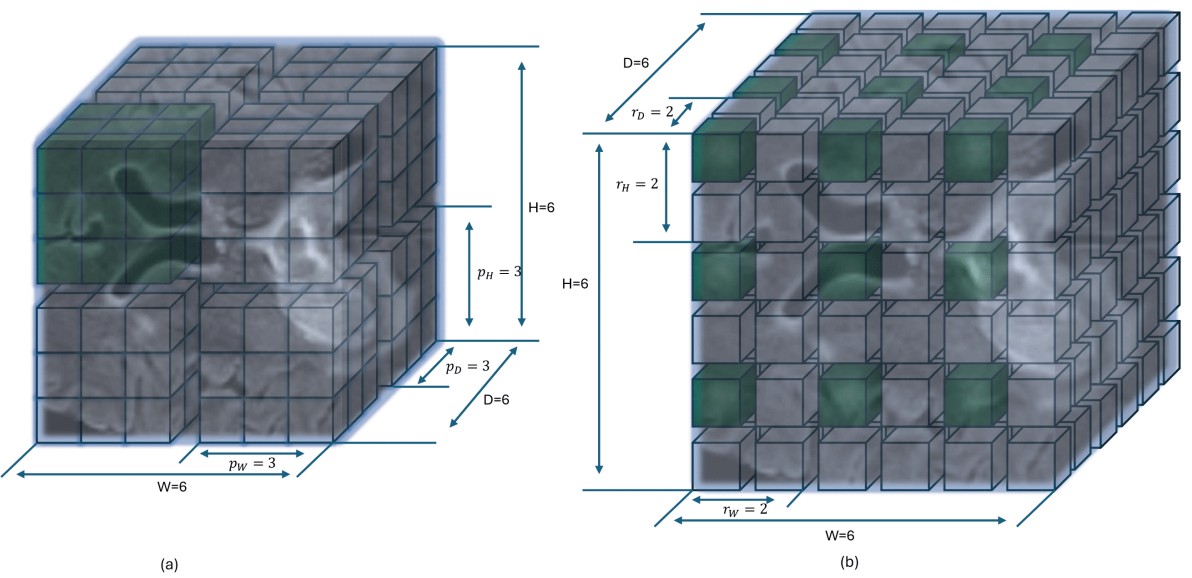

**Fig 2. (a) 3D-W-MSA computes local self-attention (highlighted in green), where the patches are adjacent to each other.** (b) 3D-DW-MSA computes global self-attention (highlighted in green). The feature map has a size of ($6 \times 6 \times 6$) and a unit size of ($3 \times 3 \times 3$) with a dilation factor $r$. Patches, represented by small cubes, are chosen at every $r^{th}$ patch across the feature map.

### 3.2.3. 3D Dilated-Window Multi-Head Self-Attention (3D-DW-MSA).

3D-DW-MSA extends the conventional 3D-SW-MSA to perform global self-attention. It incorporates dilation factors in each dimension to capture contextual information across adjacent 3D-windows in a dilated manner called the 3D dilated-window (3D-DW). 3D-DW enables the 3D-ViT block to maintain computational efficiency while capturing long-range dependencies by avoiding the dense connectivity of each patch within the attention mechanism.

To form a 3D-DW, patches from the entire feature map are selected along each dimension at intervals determined by the dilation factor $r$. The position-encoded feature map $X_{pos}$ is divided into non-overlapping 3D-DWs, with dilation factors $r_D$, $r_H$, and $r_W$ specifying the spacing between patches along the $D$, $H$, and $W$ dimensions, respectively. A 3D-DW at position $(i,j,k)$, denoted as $W_{i,j,k}$ is defined as:

$$W_{i,j,k} = X_{pos}[:,:,i \cdot r_D : i \cdot r_D + M_D, j \cdot r_H : j \cdot r_H + M_H, k \cdot r_W : k \cdot r_W + M_W]. \quad (7)$$

Where $M_D = r_D \times p_D$, $M_H = r_H \times p_H$, and $M_W = r_W \times p_W$ are the sizes of the 3D-DW along the dimensions $D$, $H$, and $W$, respectively. The same operation is repeated for all the patches in the feature map to create 3D-DWs, ensuring that all the patches across each dimension are utilized. To calculate $Q'$, $K'$, and $V'$ vectors, each 3D-DW is projected linearly as:

$$Q', K', V' = \text{Linear}(W_{i,j,k}). \quad (8)$$

Where $Q'$, $K'$, and $V' \in \mathbb{R}^{B \times M_D \times M_H \times M_W \times d}$, with $d$ dimension of each $Q'$, $K'$ and $V'$ vector. For each 3D-DW, consisting of the selected patches, the self-attention for $Q'$, $K'$, and $V'$ is computed using (4).

The total number of 3D-DW in the entire feature map, denoted by $W_{total}$ can be calculated as:

$$W_{total} = \left\lfloor \frac{D}{M_D} \right\rfloor \times \left\lfloor \frac{H}{M_H} \right\rfloor \times \left\lfloor \frac{W}{M_W} \right\rfloor. \quad (9)$$

The number of patches in each window is reduced by the dilation factor, calculated as $p_D = \frac{M_D}{r_D}$, $p_H = \frac{M_H}{r_H}$ and $p_W = \frac{M_W}{r_W}$ along the dimensions $D$, $H$, and $W$, respectively. The total number of patches in a 3D-DW is the same in numbers as in 3D-window in 3D-W-MSA, i.e., $N$. This indicates that the receptive field for self-attention is expanded globally without increasing the number of patches involved. Thus, 3D-DW-MSA maintains the same computational cost while achieving global self-attention.

Each 3D-ViT block employs two different self-attention operations, 3D-W-MSA and 3D-DW-MSA, to capture local and global features, respectively. As illustrated in Fig 3, a 3D-ViT sub-block consists of first layer normalization, multi-head self-attention (MSA), second layer normalization, and a multilayer perceptron (MLP), along with a pair of residual connections to mitigate vanishing gradients. The operation of these two consecutive 3D-ViT sub-blocks, where 3D-W-MSA and 3D-DW-MSA are applied alternatively, can be expressed as:

$$
\begin{aligned}
\hat{x}^l_{pos} &= \text{3D-W-MSA}\left(\text{LN}(x^{l-1}_{pos})\right) + x^{l-1}_{pos}, \\
x^l_{pos} &= \text{MLP}\left(\text{LN}(\hat{x}^l_{pos})\right) + \hat{x}^l_{pos}, \\
\hat{x}^{l+1}_{pos} &= \text{3D-DW-MSA}\left(\text{LN}(x^l_{pos})\right) + x^l_{pos}, \\
x^{l+1}_{pos} &= \text{MLP}\left(\text{LN}(\hat{x}^{l+1}_{pos})\right) + \hat{x}^{l+1}_{pos}.
\end{aligned}
$$

(10)

Where $\hat{x}^l_{pos}$ and $x^l_{pos}$ denote features on the 3D-W-MSA output and the corresponding MLP, respectively, while $\hat{x}^{l+1}_{pos}$ and $x^{l+1}_{pos}$ denote features on the 3D-DW-MSA output and the corresponding MLP, respectively, of the 3D-ViT block $l$.

### 3.3. 3D-ViT-UNet encoder

The 3D-ViT-UNet encoder begins with a 3D patch embedding layer, which is crucial for converting 3D volumetric data, such as 3D brain MRIs, into a sequence of patch embeddings suitable for processing by the transformer model. This layer transforms each input 3D image $X \in \mathbb{R}^{C \times D \times H \times W}$ into a high-dimensional tensor $X^{emb} \in \mathbb{R}^{\frac{D}{2} \times \frac{H}{4} \times \frac{W}{4} \times C}$, representing a low-dimensional 3D patch sequence with a patch size of $(2 \times 4 \times 4)$. This operation results in the generation of the feature maps of size $(\frac{D}{2} \times \frac{H}{4} \times \frac{W}{4})$ projected over the $C$-channel dimension and fed into the first 3D-ViT block of the encoder, illustrated in Fig 1. Positional information is incorporated into the 3D-ViT block to enhance feature maps, capturing local and global features through the DIPE, 3D-W-MSA and 3D-DW-MSA mechanisms, respectively, shown in Fig 3. Following the first 3D-ViT block, a 3D patch merging layer merges patches and prepares the feature maps for the next 3D-ViT block after performing down-sampling.

The hierarchical encoder architecture includes two additional 3D-ViT blocks, each succeeded by a 3D patch merging layer similar to the one after the first 3D-ViT block. The patch merging layer facilitates feature fusion by merging patches and concatenating feature maps from $(2 \times 2 \times 2)$ neighboring features. The total number of patches are reduced by a factor of 8 with this operation. To double the channel size, the feature channel size is reduced by a factor of 4 after each patch merging layer, with a fully connected layer. The dimensions of the resulting output feature map after successive patch merging layers in the encoder are $(\frac{D}{4} \times \frac{H}{8} \times \frac{W}{8} \times 2C)$, $(\frac{D}{8} \times \frac{H}{16} \times \frac{W}{16} \times 4C)$ and $(\frac{D}{16} \times \frac{H}{32} \times \frac{W}{32} \times 8C)$, respectively.

### 3.4. 3D-ViT-UNet bottleneck

The bottleneck serves as a critical junction, transmitting high-level features extracted from input volumetric data by the 3D-ViT-UNet encoder to the decoder. It provides a compact, information-dense feature map of 3D medical images, such as 3D brain MRIs, emphasizing essential local and global features—like tumor boundaries—while discarding irrelevant spatial details.

PLOS Digital Health

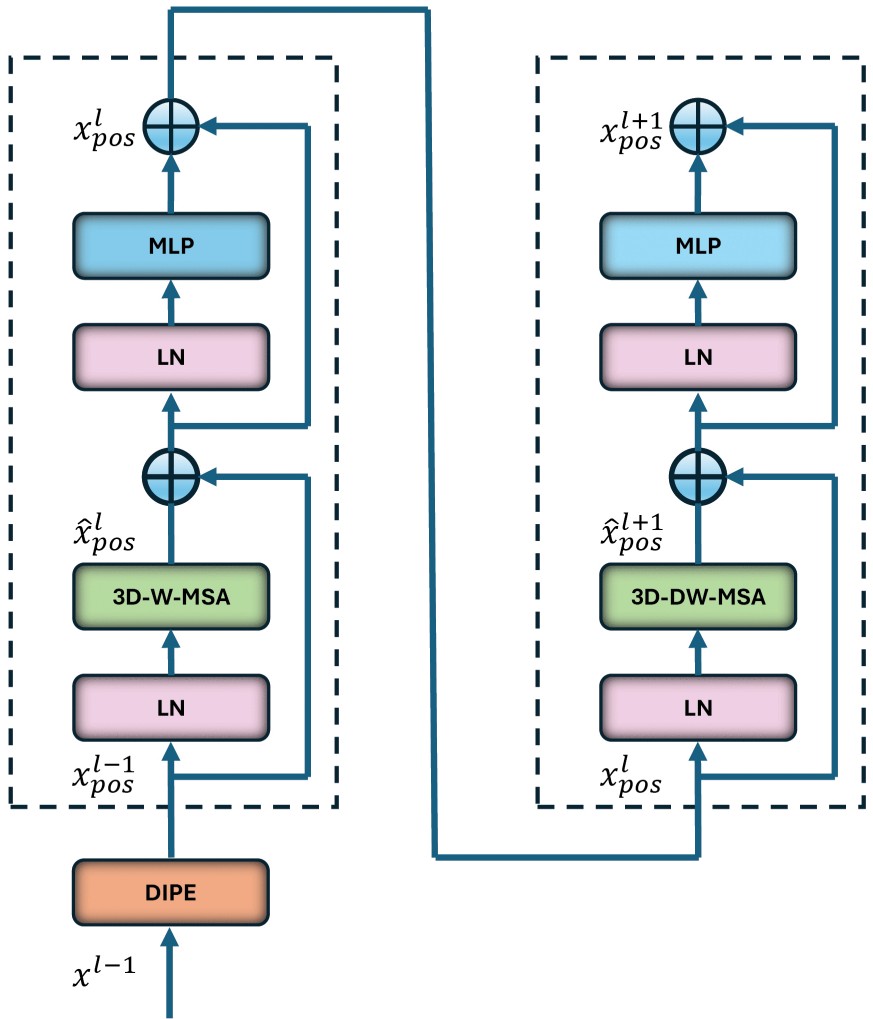

**Fig 3. A 3D-ViT block with alternating 3D-W-MSA and 3D-DW-MSA layers.**

To learn deep feature representation, the bottleneck comprises two 3D-ViT blocks, collectively containing four MSA layers: two 3D-W-MSA and two 3D-DW-MSA layers. DIPE enriches the feature maps with spatial context, while the 3D-DW-MSA layers expand the receptive field globally within the 3D-ViT blocks to capture long-range dependencies while maintaining computational efficiency. The dimensions and resolution of the features remain constant throughout the bottleneck, at $(\frac{D}{16} \times \frac{H}{32} \times \frac{W}{32} \times 8C)$.

### 3.5. 3D-ViT-UNet decoder

The 3D-ViT-UNet decoder follows a structure symmetric to the encoder, comprising three 3D-ViT blocks, each followed by a 3D patch expanding layer. Unlike the encoder, it focuses on up-sampling and reconstructing volumetric segmentation masks from the compact, low-resolution feature maps produced by the encoder. It gradually restores spatial dimensions while leveraging features extracted at high-levels by the bottleneck. In addition, the decoder integrates contextual information from each encoder layer via skip connections. It facilitates the combination of spatial details and high-level contextual features.

In order to restore spatial resolution, a reverse operation of the 3D patch merging layer is performed by each 3D patch expanding layer in the decoder. The dimensions of the bottleneck feature map are expanded by a factor of $(2 \times 2 \times 2)$ on each axis, resulting in $(\frac{D}{8} \times \frac{H}{16} \times \frac{W}{16} \times 4C)$. This up-sampling doubles the spatial dimensions while reducing channel dimension by half to enhance resolution. It retains high-dimensional features with details crucial for further processing. In the decoder, the up-sampled feature maps are then processed by the first 3D-ViT block. 3D-W-MSA and 3D-DW-MSA are employed for the extraction of local features and modeling of global dependencies effectively in the decoder's 3D-ViT block. It ensures accurate segmentation of tumor boundaries and understanding of the broader anatomical context.

The feature maps of the decoder are concatenated with the corresponding encoder's feature maps via skip connections. It helps in preserving the important spatial details lost during the down-sampling process in the encoder. This ensures the integration of high-level, context-aware features with spatially detailed features. Following the first 3D-ViT block, the decoder includes two additional 3D-ViT blocks, each preceded by a 3D patch expanding layer. The spatial resolution is restored by expanding the dimensions of the feature map by a factor of $(2 \times 2 \times 2)$, with output dimensions of $(\frac{D}{4} \times \frac{H}{8} \times \frac{W}{8} \times 2C)$, and $(\frac{D}{2} \times \frac{H}{4} \times \frac{W}{4} \times C)$, respectively, for the next two 3D-ViT blocks. The final stage of the decoder employs a 3D patch expansion layer that restores the original resolution of the feature map at the input, i.e., $(D \times H \times W)$.

Various configuration details of the proposed model are shown in Table 1.

## 4. Experiments

Dataset and evaluation, loss function, evaluation metrics, and implementation details for the proposed 3D-ViT-Unet are presented in this section.

### 4.1. Dataset and evaluation

In order to perform robust training, validation and performance comparison of the proposed model, MRI scans are needed from a diverse set of patients with detailed and heterogeneous tumor characteristics. One such choice is the Brain Tumor Segmentation 2020 (BraTS 2020) challenge dataset [53,54], which consists of multimodal 3D MRI scans.

BraTS 2020 dataset contains 369 patient cases to train and 125 patient cases to validate. It contains four different types of multimodal MRI volumes named native (T1), post-contrast T1-weighted (T1ce), T2-weighted (T2), and fluid-attenuated inversion recovery (FLAIR) [54,55]. Each modality contains volumes of size $(240 \times 240 \times 155)$, with 155 slices per volume, providing an in-depth view of both the brain and tumor regions. The annotations of the dataset comprise multiple tumor classes: enhancing tumor (ET), labeled as class 4, representing areas of active tumor cells that uptake contrast agents during imaging; peritumoral edema (ED), labeled as class 2, denoting swelling around the tumor; and the necrotic and non-enhancing tumor core (NCR/NET), labeled as class 1, indicating areas of necrotic tumor tissue. The background is labeled as class 0. For comprehensive evaluation, the aforementioned annotations are grouped into three tumor sub-regions: ET (class 4), tumor core (TC), which consists of ET (class 4) and NCR/NET (class 1), and whole tumor (WT), which combines ET (class 4), NCR/NET (class 1), and ED (class 2).

3D-ViT-UNet is trained and validated using a total of 369 BraTS 2020 training cases. The dataset is pre-processed before feeding the data into the 3D-ViT-UNet model, as it consists of multimodal MRI scans. Specifically, the three single-channel 3D modalities—T1ce, T2, and FLAIR—each with a dimension of $(240 \times 240 \times 155 \times 1)$ are combined into a three-channel volume with a shape of $(240 \times 240 \times 155 \times 3)$. Furthermore, each resulting 3D volume of size

**Table 1. Configuration Details of 3D-ViT-UNet Architecture.**

| Embed Dim | Feature Size | Number of Blocks | Window Size | Number of Heads | Params (M) | FLOPs (G) |
|---|---|---|---|---|---|---|
| 96 | 768 | [2,2,6,2] | [8,8,2] | [3,6,12,24] | 50.56 | 115.51 |

$(240 \times 240 \times 155)$ is cropped to $(128 \times 128 \times 128)$ and fed as input data for the model. Images with insufficient tumor representation are systematically excluded from the dataset. To ensure this, images are retained only if they contain at least 1% of their volume classified as tumor-related pixels, ensuring that each included image provides valuable information for model training. The retained images are then split with a ratio of 75% to train and 25% to validate the model.

## 4.2. Loss function

This study employs an integration of the focal loss function and the dice loss function for the enhancement of the segmentation capabilities of the proposed model in volumetric segmentation of brain tumors using the BraTS 2020 dataset. Focal loss is formulated specifically to address class imbalance, common among the tumor and the background classes, as well as among different tumor sub-regions, including necrosis, edema, etc.

Traditional cross-entropy loss functions tend to target well-segmented classes, which can lead to sub-optimal learning for harder-to-segment classes. In contrast, for focal loss the conventional cross-entropy loss function is modified in the case of easily-to-classify well-segmented classes by down-weighting their impact, enabling the model to target the more challenging cases, i.e., hard-to-segment classes [56]. This approach is advantageous for medical imaging in particular, where certain regions—such as brain tumors—may be under-represented in the training MRIs. The voxel-wise focal loss can be computed as:

$$L_{FL}(Y, P) = -\frac{1}{I} \sum_{i=1}^{I} \sum_{j=1}^{J} \alpha (1 - P_{ij})^{\gamma} Y_{ij} \log(P_{ij}).$$

(11)

Where $P_{ij}$ denotes the predicted probability and $Y_{ij}$ denotes the ground truth probability for class $j$ at voxel $i$. The parameters $\alpha$ and $\gamma$ control the balance and focus of the focal loss.

In order to validate medical image segmentation models, dice loss is one of the most widely used functions. It emphasizes the intersection between the predicted region and the ground truth. The voxel-wise dice loss can be calculated as:

$$L_{DL}(Y, P) = 1 - \frac{2 \sum_{i=1}^{J} \sum_{j=1}^{I} P_{ij} Y_{ij}}{\sum_{i=1}^{J} \sum_{j=1}^{I} P_{ij}^2 + \sum_{i=1}^{J} \sum_{j=1}^{I} Y_{ij}^2}.$$

(12)

It simply calculates the intersection over union (IoU) between the predicted segment and the actual class labels. It is more suitable to address the issue of class imbalance within datasets, such as BraTS 2020 [56]. Combining focal loss and dice loss results in a hybrid loss that harnesses the capabilities of both the losses. It enhances the intersection of the predicted regions with ground truth regions, expressed as:

$$L_{Hybrid}(Y, P) = \lambda_1 \cdot L_{DL}(Y, P) + \lambda_2 \cdot L_{FL}(Y, P).$$

(13)

Focal loss and dice loss are weighted with parameters $\lambda_1$ and $\lambda_2$ to focus on hard-to-segment classes, respectively. It allows the model to focus on difficult cases and enhances its performance for hard-to-segment regions. The class imbalance among the tumor classes, as well as among the sub-regions of different tumor types, is addressed by the hybrid loss effectively. The model also gains the ability of learning more generalized features by incorporating both losses, thus ensuring more robustness to class imbalance.

## 4.3. Evaluation metrics

In order to evaluate the proposed 3D-ViT-UNet for volumetric brain tumor segmentation, a broad set of quantitative metrics is used. Each evaluation metric presents a unique insight for volumetric brain tumor segmentation. These metrics

include the Dice Similarity Coefficient (DSC), 95$^{th}$ percentile of Hausdorff Distance (HD(95)), Sensitivity (SE), Specificity (SP), Precision (Pres), Normalized Surface Dice (NSD) and Mean Absolute Surface Distance (MASD).

DSC computes the intersection between the predicted segmentation and the ground truth. It measures voxel-wise similarity used to quantify the segmentation performance of a model. It is described as the fraction of twice the intersection region to the sum of the absolutes of the predicted segmentation region $P$ and the ground truth region $Y$ [24,25].

$$DSC = \frac{2|P \cap Y|}{|P| + |Y|}. \tag{14}$$

It can also be expressed as F1 score for brain tumor segmentation as:

$$DSC = \frac{2TP}{2TP + FP + FN}. \tag{15}$$

Where TP, FP and FN represent true positives, false positives, and false negatives, respectively. It makes it more sensitive to class imbalance and variations in tumor distribution.

DSC tends to increase the emphasis on the accurate detection of true tumor regions while minimizing false classifications. A very close similarity between the predicted segmentation and the ground truth is indicated by a high DSC value, underscoring the model's effectiveness.

HD(95) is another distance-based metric commonly used to evaluate the volumetric brain tumor segmentation. It finds the distance between boundaries of the predicted segmentation and ground truth. The performance of the model is evaluated in the worst cases using HD(95) by measuring how well the predicted segmentation aligns with the ground truth boundaries. An improved segmentation performance is reflected by a lower HD(95) score, indicating how accurately the model delineates tumor regions, even in challenging cases. It can be calculated as [24] as follows:

$$HD(95) = \max\{d_{95}(V_P, V_Y), d_{95}(V_Y, V_P)\}. \tag{16}$$

Where, $d_{95}(V_P, V_Y)$ and $d_{95}(V_Y, V_P)$ refer to the 95$^{th}$ percentile of the distance from the predicted segmentation voxel ($V_P$) to the ground truth voxel ($V_Y$) and the 95$^{th}$ percentile of the distance from $V_Y$ to $V_P$, respectively.

SE, also known as recall, measures the ability of the model to precisely detect positive instances, for example, the presence of a tumor. It is computed as the fraction of true positive cases identified correctly by the model. Mathematically, it is presented as follows:

$$SE = \frac{TP}{TP + FN}. \tag{17}$$

A high SE value indicates that the model effectively identifies tumor regions, which is critical for diagnosis and treatment. This means that the model is highly sensitive to identify very tiny or almost hidden tumor regions. Medical imaging becomes crucial in its role to ensure the correct recognition, identification, and detection of all possibly malignantly affected tissues. Therefore, early intervention and a better prognosis are expected for the patient.

SP measures the ratio of true negatives correctly detected by the model. It focuses on how well the model avoids false positives. It can be expressed as follows:

$$SP = \frac{TN}{TN + FP}. \tag{18}$$

SP is utilized to reduce the frequency of false alarms. In brain tumor segmentation, a high SP value ensures that non-tumor tissue is not misclassified as tumor tissue, which is critical for diagnosis and treatment. The risk of false identification of healthy tissue as tumor tissue is minimized by a high SP value to avoid unnecessary treatments or interventions. Misclassification of healthy tissue as a tumor or missing a tumor in a clinical setting can have considerably severe consequences. It highlights the significance of effective SP and SE optimization for segmentation models.

Pres is the most commonly used metric for segmentation of medical images, measuring the number of voxels classified as part of the target region that are actually correct. It highlights the model's reliability to detect positive regions and minimize over-segmentation. For brain tumor segmentation, a high pres value means that the predicted tumorous regions have a close match with the true tumor areas, excluding unnecessary background regions. It is mathematically expressed as follows:

$$Pres = \frac{TP}{TP + FP}.$$

(19)

In order to reduce over-segmentation and overestimation of tumor size, the inclusion of healthy brain tissue or surrounding edema in the segmented tumor region is avoided by high Pres value. It is crucial in medical imaging, as accurate tumor size estimation is vital for effective diagnosis, therapy planning, and tracking tumor progression or regression. High pres value results in high model reliability by reducing false positive rates for segmentation of brain tumors.

NSD evaluates how well ground truth and predicted mask boundaries align with each other by measuring overlap between the segmentation voxels of their boundaries. It is specifically helpful to assess how well the predicted surface and actual tumor boundaries align with each other to capture fine details that are essential for tumor delineation using medical imaging. A comprehensive measure of segmentation quality is ensured in terms of surface precision by incorporating both the boundary and border regions. Formally, NSD is defined as:

$$NSD(Y, P) = \frac{|B_Y \cap R_P| + |B_P \cap R_Y|}{|B_Y| + |B_P|}.$$

(20)

Where $B_Y$ and $B_P$ refer to the boundaries of the ground truth and predicted segmentation, respectively, while $R_Y$ and $R_P$ refer to the border regions of the ground truth and predicted probability within a specified tolerance. For clinical applications such as brain tumor segmentation, NSD serves as one of the most critical metrics, where boundary delineation with high precision is of key importance for accurate treatment planning and surgical interventions. A higher NSD score indicates that the predicted segmentation boundaries align well with the ground truth, signifying good model performance in capturing the tumor's surface characteristics.

MASD quantifies the mean of the average distance between the corresponding voxels on the predicted and ground truth surfaces and measures spatial dissimilarity directly. NSD focuses on boundary overlap, while MASD evaluates the geometric accuracy of the segmentation as:

$$MASD(Y, P) = \frac{1}{2} \left( \frac{\sum_{y \in Y} d(y, P)}{|Y|} + \frac{\sum_{p \in P} d(Y, p)}{|P|} \right).$$

(21)

Where $d(y,P)$ denotes the shortest distance from an instance $y$ on the ground truth surface $Y$ to the predicted voxel $P$, and vice versa for $d(p,Y)$. MASD addresses over-segmentation and under-segmentation because of its sensitivity to segmentation errors on boundaries. In order to achieve high spatial accuracy and maintain geometric fidelity for volumetric brain tumor segmentation, MASD is included in the evaluation metrics for 3D-ViT-Unet. Its lower value means well-conformed tumor boundaries.

Multiple evaluation metrics are employed in order to conduct a robust assessment of the model for segmentation. For instance, DSC may not capture boundary discrepancies, crucial for medical applications, while focusing on overlap. HD provides insight into the spatial accuracy of the model's prediction. It addresses the challenge posed by DSC. In cases of datasets with high class imbalance, such as BraTS 2020, SE and SP provide a balanced analysis on how well the model performs over different classes. DSC, SE, SP, Pres, and HD provide valuable insight into the evaluation of segmentation models for medical images. These metrics still have limitations, such as assessing boundary accuracy and spatial alignment. To address these gaps, NSD and MASD metrics were presented [57,58]. For instance, MASD is less sensitive to outliers compared to HD, offering a more stable evaluation of boundary accuracy.

### 4.4. Implementation details

To train and validate the proposed model for volumetric brain tumor segmentation, this study employs a robust configuration strategy. The experimental environment consists of Python 3.10.12 with PyTorch 2.5.0 + cu121, ensuring efficient development and deployment of the model. An NVIDIA Tesla T4 GPU with 15 GB of memory is used to accelerate deep learning model inference. This hardware setup provides computational resources to effectively process the high-dimensional MRIs.

The proposed model is trained over 140 epochs having a batch size of 2 using the Adam optimizer. The initial learning rate is set to 0.001 with a weight decay of $1 \times 10^{-5}$ to mitigate overfitting. A ReduceLROnPlateau scheduler is applied, which reduces the learning rate with a factor of 0.5 after a patience period of 5 epochs, which refines the learning rate dynamics. The combined dice and focal loss are taken advantage of by summing them to obtain better optimization.

To ensure stable training, gradient clipping is implemented in addition to the loss function to prevent exploding gradients in deep learning models. To prevent overfitting, training is halted when validation performance plateaus and a DSC-based early stopping strategy is employed. Furthermore, an L2 regularization is incorporated into the optimizer, which improves model generalization by penalizing large weighted values. Additionally, a dropout rate of 0.2 is employed within the 3D-ViT-UNet architecture, and a smoothing factor of 0.1 is applied to the loss function to further mitigate overfitting and reduce sensitivity to label noise, respectively.

## 5. Results and discussions

We have conducted experiments on BraTS 2020 to rigorously evaluate the proposed 3D-ViT-UNet in terms of segmentation performance. The experimental results are compared both quantitatively and qualitatively with the SOTA hybrid and pure transformer-based models for brain tumor segmentation.

### 5.1. Quantitative results

The proposed 3D-ViT-UNet was quantitatively evaluated using metrics such as DSC, HD(95), SE, SP, Pres, NSD and MASD. Furthermore, the proposed model is compared to the SOTA models in terms of floating-point operations (FLOPs) and the number of parameters, as shown in Table 2. It highlights the advantages of 3D-ViT-Unet in terms of computational cost and number of parameters. An input size of $(128 \times 128 \times 128)$ is used for the evaluation of all the models to ensure a fair comparison.

In terms of computational cost, 3D-ViT-Unet achieves the lowest value a 3D patient volume with 115.51G FLOPs per volume and 0.902G FLOPs per slice. With a 27.75% decrease in FLOPs per volume relative to UNETR [24], with the lowest FLOPs among the SOTA models, 3D-ViT-Unet shows a significant reduction in FLOPs compared to all SOTA models. While Swin-Unet [28] demonstrates greater parameter efficiency due to its 2D network architecture, 3D-ViT-UNet exhibits substantially lower FLOPs and superior performance across other key metrics.

Furthermore, 3D-ViT-UNet illustrates superior performance than all the SOTA models in terms of both FLOPs and the number of parameters, except Swin-Unet [28] and TansBTS [25] in terms of the number of parameters. Specifically, it

**Table 2. Performance Comparison on BraTS 2020 dataset in terms of FLOPs and number of parameters. Per Volume and Per Slice refer to the FLOPs required to segment a 3D volume and a single 2D slice of a patient's MRI, respectively.**

| Model | FLOPs (G)↓ | | Params (M)↓ |
|---|---|---|---|
| | Per Volume | Per Slice | |
| Swin-Unet [28] | 249.61 | 1.94 | 27.17 |
| TansBTS [25] | 333.07 | 2.59 | 32.99 |
| TransUnet [22] | 361.61 | 2.82 | 96.07 |
| UNETR [24] | 153.52 | 1.19 | 92.51 |
| Swin-UNETR [59] | 394.84 | 3.08 | 61.98 |
| nnFormer [51] | 421.51 | 3.29 | 149.59 |
| mmFormer [60] | 742.32 | 5.79 | 106.01 |
| CoTr [49] | 659.32 | 5.15 | 149.61 |
| UNet-Former [61] | 159.51 | 1.24 | 58.96 |
| CSWin-UNet [62] | 195.84 | 1.53 | 85 |
| **3D-ViT-UNet** | **115.51** (↓38.01) | **0.902** (↓0.287) | 50.56 |

achieves an improvement of 68.05% in FLOPs, and 47.37% in number of parameters over TransUnet [22], 72.78% in FLOPs, and 52.30% in number of parameters over nnFormer [51], 82.47% in FLOPs, and 66.46% in number of parameters over CoTr [49]. Although models like UNETR [24] and Swin-UNETR [59] have 92.5M and 61.98M parameters, respectively, fewer than other SOTA models. The proposed 3D-ViT-UNet outperforms these models' efficiency due to architectural innovations, including the attention mechanism 3D-DW-MSA. This mechanism expands the receptive field globally while maintaining the number of patches required for global self-attention, hence keeping the computational complexity low.

Fig 4 presents a detailed visual comparison of 3D-ViT-UNet with SOTA models for model size in terms of number of parameters, segmentation performance in terms of DSC and computational complexity in terms of FLOPs for BraTS 2020. TransUNet [22] and UNETR [24] have the largest model sizes and computational complexities, while their DSC scores remain sub-optimal compared to 3D-ViT-UNet. Similarly, TransBTS [25] and Swin-Unet [28] have comparatively smaller model sizes, but in terms of model complexity and DSC, these models are outperformed by 3D-ViT-UNet. 3D-ViT-UNet attains the highest DSC with the lowest computational cost, demonstrating superior performance in comparison to all SOTA models as depicted in Fig 4. These findings reinforce the significance of efficient design choices such as 3D-W-MSA, 3D-DW-MSA, and DIPE.

The above results illustrate how effective the proposed 3D-ViT-UNet is in terms of computational efficiency, suggesting that the model is optimally suited for biomedical applications and can be deployed in healthcare systems using limited computational and memory resources.

The Four-fold cross-validation results of 3D-ViT-UNet for DSC and HD(95) are depicted in Table 3. 3D-ViT-UNet showcases competitive performance across all folds and for all tumor sub-regions. It achieves the highest DSC of 91.53% for WT in Fold 2, 82.91% for TC in Fold 3, and 80.61% for ET in Fold 2. The average DSC in all folds is 90.73% for WT, 81.40% for TC, and 79.76% for ET, with an overall mean DSC of 83.96%.

The proposed model also attains an average HD(95) of 4.18 mm for WT, 4.75 mm for TC, and 5.51 mm for ET, which results in an overall average HD(95) of 4.81 mm. It demonstrates the capabilities of the model to segment tumor boundaries precisely. In Fold 2, the lowest HD(95) value of 4.10 mm for WT further emphasizes the robustness of the model to accurately delineate WT boundaries.

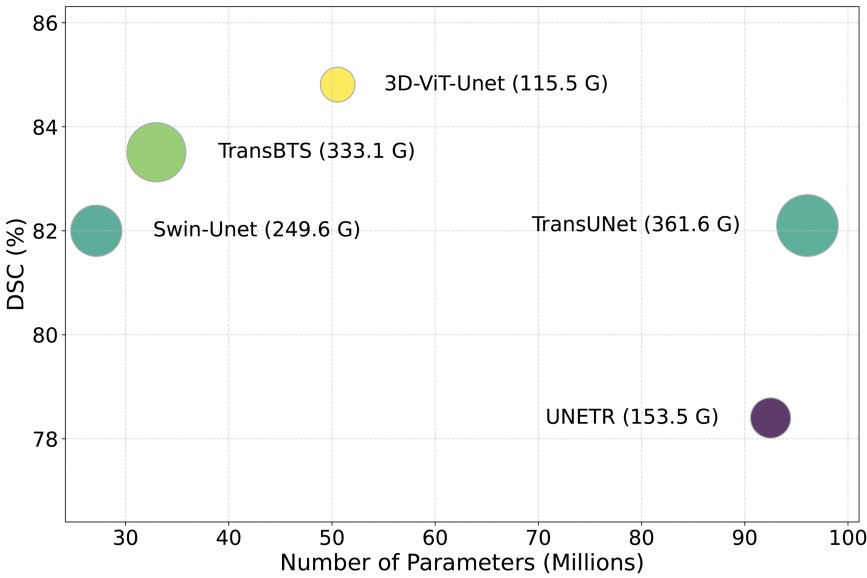

**Fig 4. The number of parameters in millions vs DSC is presented in the figure, where the computational complexity by the bubble size indicates FLOPs.** Compared to SOTA models, 3D-ViT-UNet achieves highest DSC value with a smaller model size and lower computational complexity.

**Table 3. Four-Fold cross-validation of 3D-ViT-UNet for DSC and HD(95) on ET, TC and WT.**

| Metric | DSC% ↑ | | | | HD(95) ↓ | | | |
|---|---|---|---|---|---|---|---|---|
| Tumor | TC | WT | ET | Avg. | TC | WT | ET | Avg. |
| Fold 1 | 80.41 | 90.54 | 78.32 | 83.09 | 4.91 | 4.23 | 5.35 | 4.83 |
| Fold 2 | 82.67 | **91.53** | **80.61** | **84.94** | 4.87 | 4.12 | **5.26** | **4.75** |
| Fold 3 | **82.91** | 91.43 | 79.96 | 84.76 | 4.63 | **4.10** | 5.61 | 4.78 |
| Fold 4 | 79.63 | 89.42 | 80.18 | 83.07 | **4.61** | 4.29 | 5.83 | 4.91 |
| **Avg.** | 81.40 | **90.73** | 79.76 | 83.96 | 4.75 | **4.18** | 5.51 | 4.81 |

In terms of DSC and HD(95), the proposed 3D-ViT-UNet achieves improved segmentation performance compared to SOTA models due to its end-to-end volumetric segmentation framework built on an advanced 3D-ViT architecture consisting of an effective positional encoding strategy known as DIPE. Thus, the sequence of input patches is maintained all over the network, which is of key importance for tumor segmentation, highly dependent on adjacent tissues.

Table 4 illustrates the comparative analysis of 3D-ViT-UNet with respect to the SOTA models on the BraTS 2020 dataset, highlighting its superior performance for segmentation of brain tumors, specifically in terms of DSC and HD(95). Leveraging a ViT-based architecture, the 3D-ViT-UNet consistently outperforms or competes closely with existing techniques across tumor sub-regions. With an average DSC of 84.81%, the model achieves a notable improvement over other methods, illustrating its robustness to accurate brain tumor segmentation. 3D-ViT-UNet achieves a DSC of 91.03% for WT, outperforming SOTA models such as Swin-Unet [28], UNETR [24], TransBTS [25], UNETR++ [30], TMA-TransBTS [31] and AST-Net [63] with a significant margin. This performance of the model shows its ability to capture local and global features to segment WT. 3D-ViT-Unet shows a notable improvement over Swin-BTS [64] and other SOTA models. The model achieves a competitive DSC of 84.81% outperforming AST-Net, which holds the highest DSC among the SOTA models. Overall, 3D-ViT-UNet demonstrates the highest average DSC across all tumor sub-regions.

**Table 4. Comparison on BraTS 2020 dataset, 3D-ViT-UNet achieves better segmentation results in terms of DSC and HD(95).**

| Model | DSC% ↑ | | | | HD(95) ↓ | | | |
|---|---|---|---|---|---|---|---|---|
| | TC | WT | ET | Avg. | TC | WT | ET | Avg. |
| Swin-Unet [28] | 77.11 | 89.58 | 79.21 | 81.96 | 14.58 | 7.91 | 11.01 | 11.17 |
| TransBTS [25] | 81.53 | 90.09 | 78.93 | 83.51 | 9.76 | 4.96 | 17.95 | 10.89 |
| TransUnet [22] | 78.38 | 89.51 | 78.42 | 82.10 | 12.76 | 6.12 | 12.93 | 10.60 |
| UNETR [24] | 76.79 | 88.13 | 70.31 | 78.44 | 10.63 | 8.18 | 34.96 | 17.75 |
| Swin-BTS [64] | 79.28 | 89.19 | 78.25 | 82.24 | 15.78 | 8.56 | 26.84 | 17.06 |
| VT-Unet [65] | 79.28 | 88.84 | 77.45 | 81.86 | 14.76 | 9.54 | 28.99 | 17.76 |
| UNETR++ [30] | 78.49 | 89.03 | 75.30 | 80.94 | 8.019 | 6.209 | 6.019 | 6.749 |
| TMA-TransBTS [31] | 79.64 | 90.31 | 76.86 | 82.27 | 6.603 | 4.537 | 5.904 | 5.68 |
| AST-Net [63] | **84.10** | 90.29 | 77.71 | 84.03 | 6.12 | 6.05 | 30.43 | 14.20 |
| **3D-ViT-UNet** | 82.49 | **91.03** | **80.91** | **84.81(↑0.78)** | **4.65** | **4.23** | **5.74** | **4.87(↓0.81)** |

The 3D-ViT-UNet achieves an average HD(95) of 4.87 mm, illustrating its high ability to capture tumor boundaries more precisely than the SOTA methods. The model achieves an HD(95) of 4.23 mm for WT, outperforming Swin-Unet [28] and TransUnet [22] by 3.67 mm and 1.77 mm, respectively, further highlighting its capabilities of capturing accurate global features for WT segmentation. For ET, the proposed model achieves a HD(95) of 5.74 mm, showing improvement over TMA-TransBTS [31], which has an HD(95) of 5.904 mm, the lowest among SOTA methods. The model also records an HD(95) of 4.65 mm for TC, demonstrating an improvement over TransBTS [25], AST-Net [63], and TMA-TransBTS [31], marking the smallest value among the SOTA models.

The 3D-ViT-UNet demonstrates improved segmentation accuracy and boundary precision, achieving optimal DSC and HD(95), making it suitable for clinical applications.

Further quantitative evaluation and comparison of 3D-ViT-UNet with SOTA models in terms of SE, SP, and Pres on the BraTS 2020 dataset have been summarized by Table 5. 3D-ViT-UNet achieves an average SE of 79.8%, with a significant performance for WT of 91.2%, outperforming all competing models, including HUT [66] with an SE score of 90.3%. However, for TC and ET regions, 3D-ViT-UNet achieves an SE of 78.1% and 70.1%, respectively, slightly lower than HUT's [66] SE of 81.2% and 77.2%. This suggests the potential challenges in detecting smaller tumor sub-regions with subtle features like ET, a common limitation across competing models.

3D-ViT-UNet demonstrates persistent performance in the three tumor regions with an average SP value as high as 96.6%. High SP values depict the sensitivity and a balanced approach of the 3D-ViT-UNet to avoid over-segmentation while detecting true positives. This improves the performance of the model in terms of reducing the false positives, hence enabling the model to accurately exclude the non-tumor regions.

3D-ViT-UNet outperforms all SOTA models in terms of Pres with a value of 92.6% for WT, such as HUT [66] and TransBTS [25], with Pres values of 87.3% and 91.2%, respectively. For TC, the Pres score of 88.3% is competitive,

**Table 5. Comparison on BraTS 2020 dataset, 3D-ViT-UNet achieves better segmentation results in terms of SE, SP and Pres.**

| Model | SE↑ | | | | SP ↑ | | | | Pres ↑ | | | |
|---|---|---|---|---|---|---|---|---|---|---|---|---|
| | TC | WT | ET | Avg. | TC | WT | ET | Avg. | TC | WT | ET | Avg. |
| Swin-Unet [28] | 0.801 | 0.892 | 0.725 | 0.806 | — | — | — | — | 0.771 | 0.832 | 0.735 | 0.779 |
| TransBTS [25] | 0.751 | 0.837 | 0.641 | 0.743 | — | — | — | — | **0.884** | 0.912 | 0.841 | 0.879 |
| HUT [66] | **0.812** | 0.903 | **0.772** | **0.829** | — | — | — | — | 0.824 | 0.873 | 0.782 | 0.826 |
| 3D-ViT-UNet | 0.781 | **0.912** | 0.701 | 0.798 | 0.968 | 0.981 | 0.949 | 0.966 | 0.883 | **0.926** | **0.852** | **0.887(↑0.008)** |

though slightly lower than TransBTS [25], which attains 88.4%. However, 3D-ViT-UNet achieves a Pres score of 85.2%, higher than TransBTS [25] and HUT [66] with Pres of 84.1% and 78.2%, respectively, showcasing the ability of the model to effectively perform the challenging task of segmenting ET. This high-precision value averaging at 88.7% demonstrates the model's ability of maintaining lower false-positive rates, particularly in the WT region, which has clearer boundaries than the more complex and smaller ET regions.

The performance of 3D-ViT-UNet in terms of NSD and MASD for BraTS 2020 dataset, as presented in Table 6, underscores its superiority in delineating tumor boundaries with robustness and precision compared to SOTA models. 3D-ViT-Unet achieves an NSD value of 42.36. It outperforms Swin-UNETR by 3.03% improvement in NSD [59]. The model attains the highest NSD values of 48.2, 40.8, and 38.1 for WT, TC, and ET, respectively, showing its consistent superiority over SOTA models. It highlights the effectiveness of the model in capturing global context while preserving fine-grained details due to its advanced attention mechanism. Similarly, 3D-ViT-UNet achieves an average MASD of 0.89 mm, which is significantly lower than nnFormer [51] (1.67mm) and TransBTS [25] (1.91mm), underscoring its precision in surface localization. The model's performance in individual sub-regions, i.e., WT, TC, and ET, further demonstrates its superiority over nnFormer, with an improvement of 57.84%, 31%, and 41.4%, respectively. The aforementioned results in terms of NSD and MASD prove the suitability of the proposed models for clinical applications, with the requirement of high accuracy segmentation at tumor boundaries.

### 5.2 Qualitative results

The qualitative comparison of the proposed 3D-ViT-UNet with SOTA models is presented in Fig 5, highlighting over-segmentation, under-segmentation, and blending of tumor sub-regions. As evident from the red- and blue-circled regions, 3D-ViT-UNet demonstrates superior performance in boundary delineation. The predictions of the proposed model are closely aligned with the ground truth. Additionally, it minimizes blending into adjacent non-tumor regions, even in complex boundary scenarios. 3D-ViT-UNet captures irregular shapes of the ET while maintaining clear separation between tumor sub-regions (NCR/NET, ED, and ET), effectively.

In contrast to 3D-ViT-UNet, TransBTS faces significant challenges in delineating sharp boundaries. In particular, it tends to blend the ET into the neighboring ED region, as highlighted in the red-circled region, while failing to accurately capture the boundaries of NCR/NET, as shown in the blue-circled area. Similarly, Swin-Unet and Swin-BTS exhibit inadequate boundary localization in the blue region, particularly, imprecisely modeling the transitions between ED and NCR/NET, leading to misclassified areas.

3D-ViT-UNet, on the other hand, demonstrates its good performance in minimizing both over-segmentation and under-segmentation, as shown in blue-circled region, where the model accurately localizes the tumor component without extending boundaries into surrounding healthy tissues. In comparison, SOTA models such as UNETR struggle with under-segmentation due to excessive smoothing of NCR/NET boundaries, as evident in the blue- and red-circled regions. This results in a loss of distinction between ED and NCR/NET. Swin-Unet and TransUnet also exhibit under-segmentation, where critical portions of ET and NCR/NET are omitted, as highlighted in the red- and blue-circled regions, respectively.

**Table 6. Comparison on BraTS 2020 dataset, 3D-ViT-UNet achieves better segmentation results in terms of NSD and MASD.**

| Model | NSD ↑ | | | | MASD ↓ | | | |
|---|---|---|---|---|---|---|---|---|
| | TC | WT | ET | Avg. | TC | WT | ET | Avg. |
| TransBTS [25] | 37.48 | 46.09 | 33.61 | 39.06 | 1.39 | 2.48 | 1.86 | 1.91 |
| UNETR [24] | 37.67 | 41.31 | 35.20 | 38.06 | 1.29 | 2.25 | 2.11 | 1.88 |
| nnFormer [51] | 38.17 | 40.22 | 34.41 | 37.60 | 1.41 | 2.04 | 1.57 | 1.67 |
| Swin-UNETR [59] | 38.31 | 43.89 | 35.80 | 39.33 | 1.31 | 2.34 | 2.23 | 1.96 |
| **3D-ViT-UNet** | **40.79** | **48.21** | **38.10** | **42.36 (↑3.03)** | **0.89** | **0.86** | **0.92** | **0.89 (↓0.78)** |

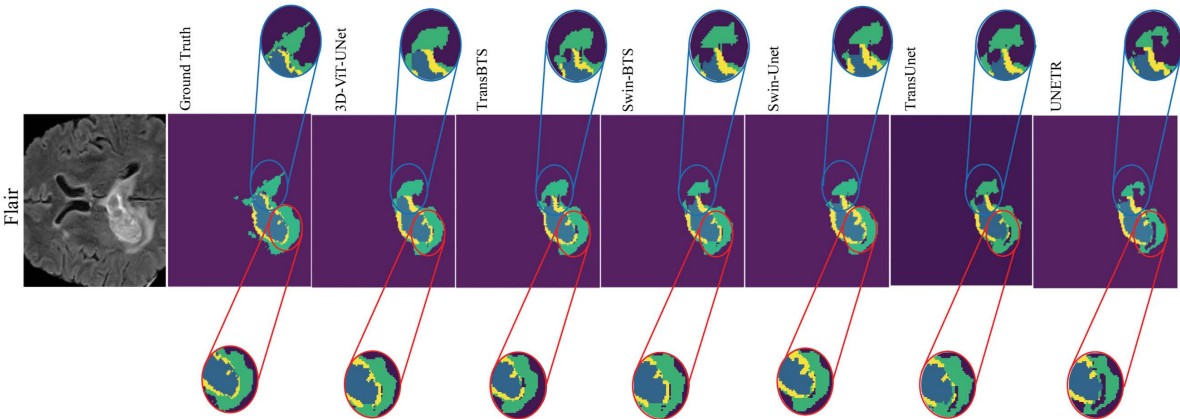

**Fig 5. Illustrates over-segmentation, under-segmentation, blending of regions, and boundary delineation issues.** The colors yellow, green, and blue represent the enhancing tumor (ET), edema (ED), and necrotic/non-enhancing tumor (NCR/NET) regions, respectively.

It is evident from the qualitative results in Fig 5 that TransBTS and Swin-BTS demonstrate misclassification, with the red-circled region highlighting nearby structures incorrectly labeled as part of ET, thereby compromising segmentation accuracy. The blue-circled region illustrates boundary localization inaccuracies by TransBTS, where ET and NCR/NET regions are merged with blurred and imprecise boundaries. Swin-Unet and TransUnet face similar issues, with noticeable boundary localization errors in both the red- and blue-circled regions. As a result, segmentation accuracy is reduced for complex tumor cases.

The tendency to over-smooth, shown in the blue-circled region for models such as UNETR and Swin-BTS, is one of their most crucial shortcomings. It causes the blending of NCR/NET and ED, leading to a failure to maintain sharp distinctions. This over-smoothing effect degrades segmentation accuracy by masking critical tumor details, ultimately affecting the model's clinical reliability.

Maintaining the structural integrity of tumorous regions is crucial, which is achieved by accurate segmentation of fine details. Fig 6 focuses on the model's ability to segment small NCR/NET regions within the ET region (highlighted in the yellow-circled area) and the background details. It also demonstrates the model's capability to segment fine-grained NCR/NET components (highlighted in the cyan-circled region).

The proposed model accurately identifies small NCR/NET areas, as shown in both the yellow- and cyan-circled regions. The predictions are closely aligns with the ground truth within the ET region. It results in an enhanced sensitivity of the model to capture very fine intra-tumoral variations. Similarly, the model effectively segments the background details and delineates fine-grained NCR/NET components. It preserves the structural nuances of these regions (see the cyan-circled area). This reflects the model's superior performance and ability to capture both local and global spatial dependencies. Highlighted by the yellow- and cyan-circled regions, a significant limitation can be observed while capturing fine-grained details in qualitative results of the SOTA models. For small areas like NCR/NET inside the ET region, models like TransBTS and Swin-BTS cause boundary inaccuracy and under-segmentation, resulting in blending with the surrounding structures. Furthermore, over-segmentation and blur transitions can be observed in the visual results of both the models while separating NCR/NET boundaries from the background, degrading the segmentation precision. Missclassification of NCR/NET and the background region is observed for TransUnet and Swin-Unet, which shows that they fail to resolve complex spatial transitions highlighted by the cyan-circled region. Over-smoothing reduces the structural fidelity in the ET region as these models are unable to capture very fine NCR/NET components, as shown in the yellow-circled region. Structural reliability and segmentation accuracy are further degraded in case of UNETR as it causes over-smoothing and over-segmentation shown in the yellow- and cyan-circled regions.

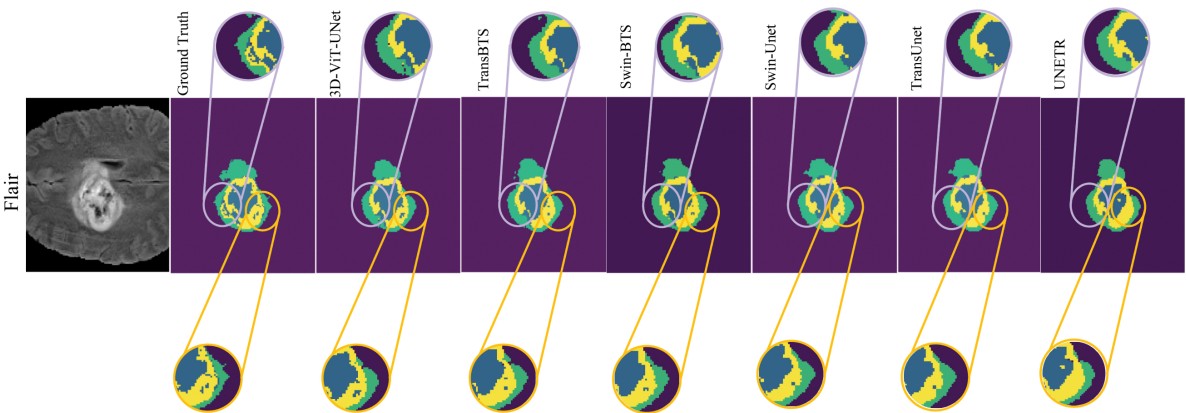

**Fig 6. Illustrates the model's ability to capture fine-grained details in the segmentation.**

In terms of structural integrity preservation and resolution of complex morphology of tumors, the segmentation acuracy of the proposed 3D-ViT-UNet for fine-grained features shows the superiority of the model. Based on the qualitative results described above, the proposed 3D-ViT-UNet can be considered as one of the most reliable methods for tumor characterization with better accuracy than the existing SOTA methods.

In order to maintain the structural integrity of the tumorous regions, a qualitative comparison of 3D-ViT-UNet with SOTA models in terms of shape preservation is shown in Fig 7. 3D-ViT-UNet demonstrates better shape preservation, as highlighted in black-circled region, where it maintains the irregular contours of NCR/NET without over-segmentation and shape distortion, closely aligned with the ground truth. Additionally, in light blue-circled area, our proposed model accurately preserves the finer structural details and shapes of NCR/NET, ensuring a prominent distinction from the surrounding ET and ED regions.

However, SOTA models exhibit notable limitations in shape preservation. TansBTS for instance, struggles with over-segmentation, resulting in distorted NCR/NET boundaries, shape, and deviation from the ground truth, as shown in black-circled region. As a result of blending in NCR/NET and ED regions the loss of shape and structural details is further shown in the light blue-circled region. Similarly, Swin-BTS, while performing moderately, introduces boundary inaccuracies and over-smoothing in NCR/NET shapes (highlighted in the black- and light blue-circled regions, respectively).

Due to blending and over-smoothing, models such as Swin-Unet and TransUnet are unable to preserve the shape of NCR/NET regions. As a result, under-segmentation affects the precision of the model. UNETR also performs poorly, exhibiting excessive over-smoothing that erases NCR/NET boundaries in the black-circled region and merges NCR/NET with ED in the light blue-circled region, creating artifacts and severely compromising segmentation accuracy. These limitations highlight the critical importance of shape preservation in brain tumor segmentation. Hence, 3D-ViT-UNet demonstrates its ability to ensure an accurate tumor shape characterization. As a result, it reinforces the reliability and clinical potential of the model for precise tumor analysis. Therefore, it sets 3D-ViT-Unet apart from the existing SOTA models.

Fig 8 presents a visual comparison of the proposed model with SOTA models for clinically meaningful class-wise semantic segmentation. 3D-ViT-UNet competes with SOTA models, addressing common limitations by maintaining a clear delineation among all the classes. For instance, UNETR and Swin-BTS exhibit poor class separation due to over-smoothing. 3D-ViT-Unet prevents the blending of class 2 and class 4. It also reduces the blending of class 0 into adjacent classes. 3D-ViT-UNet outperforms SOTA models in terms of under-segmentation and misclassification. It accurately preserves the boundaries of class 2, demonstrating the best segmentation precision. For class 4, which is particularly challenging to segment, our proposed model maintains structural integrity, minimizes boundary inaccuracies and blending

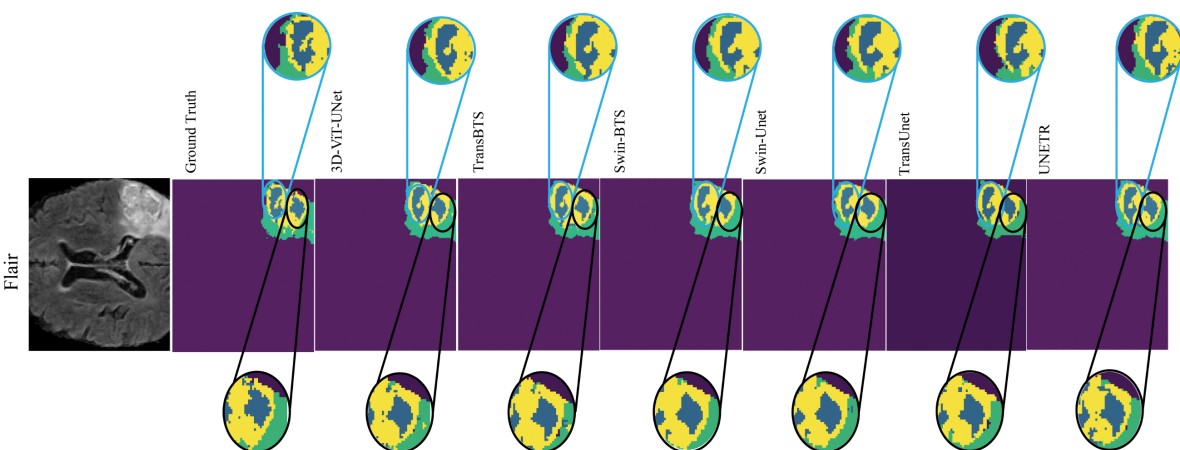

**Fig 7. Highlights the models' capabilities in preserving the shape and structural integrity of the segmented regions.**

issues. TransBTS faces the limitation of under-segmentation, and Swin-BTS struggles with blending issues. Hence, qualitative analysis of the class-wise semantic segmentation in Fig 8 reinforces the findings of the qualitative results in Fig 5. These visual results showcase the superior performance of 3D-ViT-UNet to address key segmentation challenges and its clinical potential for accurate brain tumor segmentation.

A comparative class-wise evaluation of the proposed model with the SOTA models is presented in Fig 9. These results highlight enhanced capabilities of the proposed model in terms of class separability, boundary localization, and structural preservation. It analyzes the mitigation of over-segmentation, under-segmentation, and blending. Fig 9 shows that the proposed model performs better at reducing over-segmentation compared to other SOTA models, particularly in class 4. Tumor boundaries are preserved and misclassification of non-tumor regions as part of the lesion is prevented. 3D-ViT-UNet accurately localizes Class 4 compared to SOTA models; hence, it minimizes the excessive inclusion of surrounding healthy tissues. It precisely differentiates tumor regions from the background, thereby reducing false-positive rates due to its end-to-end 3D-ViT-based architecture, which enhances spatial awareness.

SOTA models such as Swin-Unet extend tumor regions beyond their actual boundaries, especially in class 2 and class 4, resulting in over-segmentation. These models struggle to maintain a clear separation between tumor and non-tumor tissues, leading to a loss of specificity in classification. Similarly, TransBTS and Swin-BTS demonstrate moderate over-segmentation, but their errors remain prominent in class 2 and class 4.

The proposed 3D-ViT-UNet improves the mitigation of under-segmentation of class 1 and class 4. Leveraging its advanced attention mechanism, the model prevents the omission of clinically significant structures and captures fine-grained tumor components with high-precision. The model precisely identifies and preserves small components of class 1. It also accurately segments class 4, critical for clinical treatment. Other SOTA methods, such as TransBTS and Swin-BTS under-segment for class 1, class 2, and class 4. Swin-Unet and TransUnet excessively omit crucial tumor components and leading to under-segmentation. It causes segmentation inconsistency. UNETR faces severe under-segmentation of class 2 and class 4 due to the loss of key portions.

In terms of boundary localization and preservation of sharp transitions, the proposed model demonstrates comparatively better performance, as shown in 9. Leveraging, 3D-DW-MSA to compute global spatial relationships, the model enhances boundary separability, ensuring clear tumor differentiation. This capability is important for accurate tumor grading and treatment planning. Compared with other SOTA models such as TransBTS and Swin-BTS which merge class 1 and class 2 into surrounding regions, suffering from boundary blending and reduced classification specificity. Swin-Unet

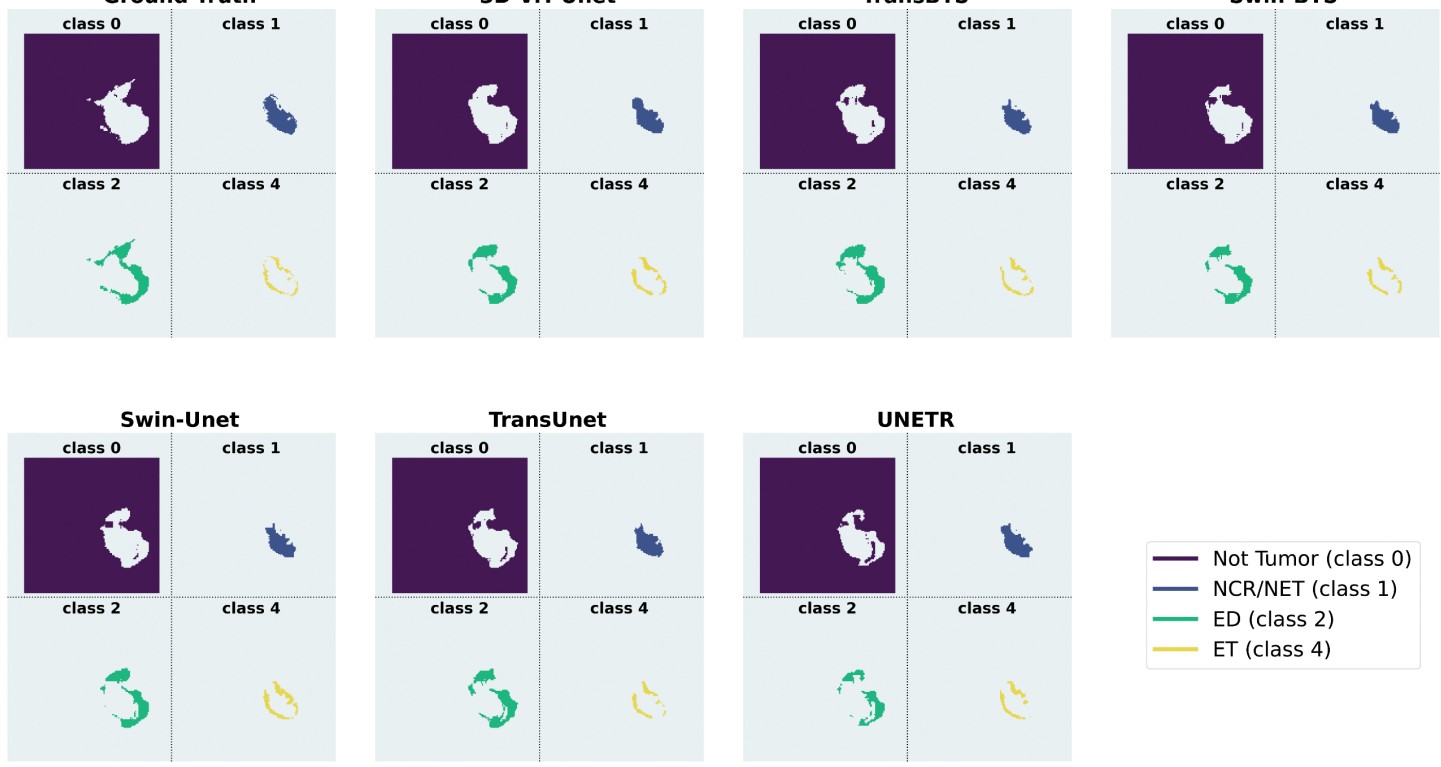

**Fig 8. Qualitative results for class-wise semantic segmentation.**

and TransUnet struggle to maintain margins for class 1, leading to classification inconsistencies. Due to the loss of critical segmentation details, UNETR blends tumor boundaries in class 2 and class 4.

The qualitative results of 3D-ViT-UNet presented in Fig 9 reinforce its superiority in mitigating over-segmentation, under-segmentation and blending. These results align closely with the findings depicted in Figs 5 and 8, further validating the robustness of the proposed 3D-ViT-UNet as the most clinically reliable model for brain tumor segmentation.

## 6. Conclusion

3D-ViT-UNet is a pure ViT-based architecture proposed for end-to-end volumetric brain tumor segmentation. It integrates advanced mechanisms such as 3D-W-MSA, 3D-DW-MSA, and DIPE to effectively address key challenges in volumetric segmentation, including the preservation of spatial dependencies and computational efficiency. Extensive evaluations of the BraTS 2020 dataset confirm that 3D-ViT-UNet achieves superior segmentation accuracy, particularly in complex tumor regions such as Enhancing Tumor (ET), Peritumoral Edema (ED), and Necrotic/Non-Enhancing Tumor Core (NCR/NET). It outperforms SOTA methods, including Swin-Unet, TransBTS, UNETR, UNETR++ and TMA-TransBTS as quantitatively validated by a significant improvement in DSC, HD95, and other metrics.

The proposed model also delineates complex tumor structures with enhanced accuracy, as illustrated by the qualitative results. It effectively mitigates common challenges such as over-segmentation, under-segmentation, and blending, which often lead to misclassification of tumor classes. The model excels in balancing high performance with reduced computational cost, as demonstrated by its smaller parameter count and FLOPs, achieving a DSC of 84.81% while maintaining robustness across tumor sub-regions. The qualitative results align with the quantitative results, further validating the reliability of 3D-ViT-UNet for clinical deployment.

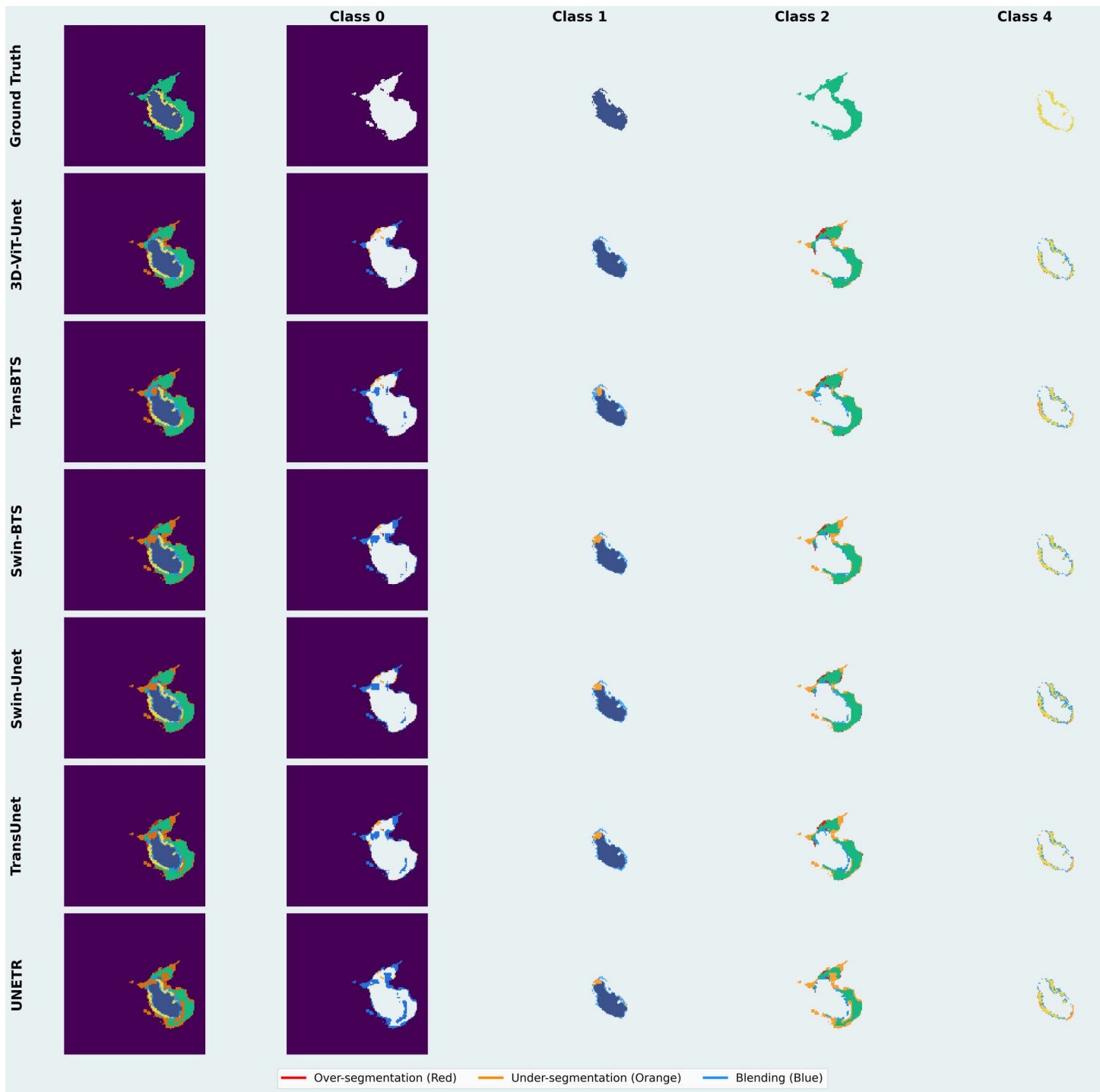

**Fig 9. Qualitative evaluation of class-wise segmentation in terms of over-segmentation, under-segmentation and blending.**

## Author contributions

**Conceptualization:** Sikandar Afridi, Atif Jan, Muhammad Irfan Khattak.

**Data curation:** Sikandar Afridi, Taimur Ahmed Khan.

**Formal analysis:** Sikandar Afridi.

**Investigation:** Sikandar Afridi.

**Methodology:** Sikandar Afridi, Atif Jan, MUHAMMAD ABEER IRFAN, Taimur Ahmed Khan.

**Supervision:** Atif Jan, MUHAMMAD ABEER IRFAN, Muhammad Irfan Khattak.

**Validation:** Atif Jan, Taimur Ahmed Khan.

**Visualization:** MUHAMMAD ABEER IRFAN.

**Writing – original draft:** Sikandar Afridi.

**Writing – review & editing:** MUHAMMAD ABEER IRFAN.

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
